# Generating Robot Capability Maps
# with Neural Fields

Xinyu Chen
Technical University of Munich
xinyu-lucy.chen@tum.de

Jonathan Külz
Technical University of Munich
jonathan.kuelz@tum.de

Matthias Althoff
Technical University of Munich
althoff@tum.de

*Abstract*—Many robotic applications, such as manipulation and human-robot interaction, require accurate knowledge about the workspace of a manipulator. Further, an abstraction of the capabilities of a robot arm within its workspace, often modeled by so-called capability maps, is important for grasp and task planning. Unfortunately, existing methods to identify the capabilities of a manipulator are time-consuming and data-intensive. This work proposes generating robot capability maps directly in task space leveraging neural fields trained entirely on synthetically generated data. In numerical experiments, we show that our approach generalizes over various morphologies and produces accurate capability maps within milliseconds.

## I. INTRODUCTION

The workspace of a manipulator – defined as the set of all positions it can reach – is an essential aspect in determining its operational suitability for a given task. For many applications, it is interesting to generate so-called capability maps by assigning a capability measure to every position in the workspace, such as the manipulability index proposed by Yoshikawa [29]. Capability maps contribute to many subsequent tasks, such as motion planning [30, 20, 24], localization [21, 23], human-robot interaction [25, 31], and hardware design [14]. Generating accurate capability maps with traditional approaches takes hours to compute [30, 20]. Especially in scenarios where the robot morphology is subject to change, such as in the context of modular robots [27, 1] or robot design [7, 12], the computational complexity thus drastically restricts the applicability of capability maps.

This work uses neural fields [26] to efficiently generate capability maps for diverse serial manipulators. In numerical experiments, we show that accurate capability maps with more than 300,000 query positions can be created in less than 0.05s on average. Furthermore, we show that our approach generalizes to out-of-distribution samples.

## II. RELATED WORK

To generate capability maps, conventional approaches discretize the task space, estimate the reachability of each discrete position, and compute a capability measure whenever a position is reachable. This usually requires solving the inverse kinematics repeatedly, resulting in long runtimes [30, 25, 14, 24]. Porges et al. [20] propose a more efficient method to generate capability maps by sampling random joint angles, computing the forward kinematics, and eventually refining the capability maps by inverse kinematics computations. However,

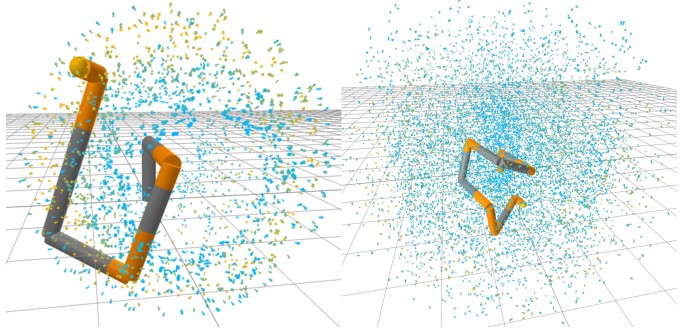

Fig. 1. Two modular robots with three (left) and six (right) degrees of freedom and samples from their workspaces. Link modules are shown in grey, and joint modules are shown in orange. The colored dots represent end effector poses sampled by computing the forward kinematics for random joint angles. Blue dots indicate a low manipulability measure, while the orange ones represent positions at which the robot is locally more capable.

following their approach, even the coarsest map takes 15 seconds to generate, while more precise maps can take a couple of minutes to more than a day.

One application of capability maps is the efficient generation of heuristics to evaluate the performance of different robot morphologies, e.g., to optimize compositions of modular reconfigurable robots [4, 28, 17]. However, this often requires evaluating millions of robots [16], rendering the aforementioned methods for capability map computation infeasible. Jamone et al. [10] and Boanta and Brisan [2] use neural networks to estimate the workspace of a robot. However, the work of Jamone et al. [10] assumes that the robot has a fixed morphology, while the method of Boanta and Brisan [2] can not capture the workspace geometry and only predicts the workspace volume.

Neural Fields [26] have recently attracted significant attention as representations for temporal and spatial data. Existing works use neural fields to represent scenes or distance-to-collision functions in robot joint space [11, 15, 22]. Related to our approach, Park et al. [19] use a signed distance function to represent various shapes, while Mescheder et al. [18] train a binary occupancy classifier. Both works create a conditional neural field [26] to reconstruct various 3D shapes by providing a latent vector that represents object features.

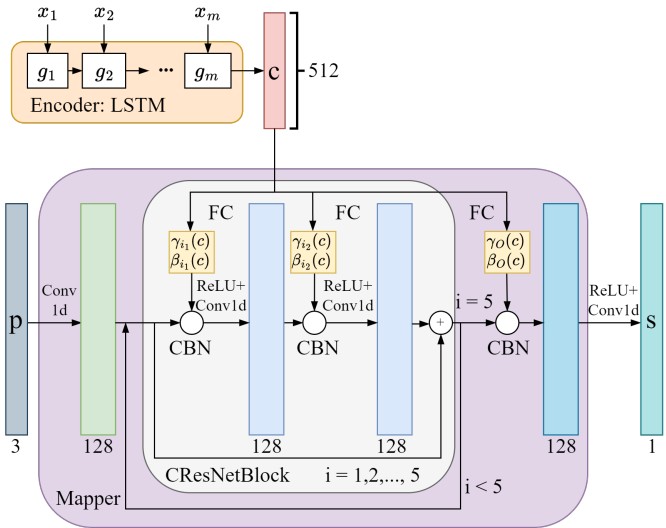

Fig. 2. Model architecture: The orange block represents an LSTM embedding robot features in a latent vector $c$. The purple block represents the mapper that predicts the capability measure $s$ based on query position $p$. The normalization parameters $\gamma_i$ and $\beta_i$ (yellow) for the CBN in the CResNetBlocks are computed by passing the latent vector $c$ to fully connected layers. There are five CResNetBlocks in the model.

## III. APPROACH

We define the positional workspace $\mathcal{W} \subset \mathbb{R}^3$ of a manipulator as the set of all end effector positions over all possible joint angles in its joint space $\mathcal{Q}$. For a capability function $m : \mathcal{Q} \to \mathbb{R}$, the manipulator capability map $f : \mathbb{R}^3 \to \mathbb{R}$ is

$$
f(p) = \begin{cases} \max_{\mathrm{FK}_p(q)=p} m(q) & , \text{ if } p \in \mathcal{W} \\ \nu & , \text{ otherwise} , \end{cases} \tag{1}
$$

where $\mathrm{FK}_p(\cdot)$ is the positional forward kinematics, i.e., the position of the end effector for joint angles $q$. The special value $\nu \in \mathbb{R}$ indicates that position $p$ is not reachable and can be chosen according to the used capability function. In this work, we use a conditional neural field to learn an approximation of the capability map, given an encoding of the robot morphology.

### A. Learning Pipeline

Figure 2 shows our model architecture. We encode modular robots composed of $m$ modules in an $n$-dimensional vector per module and stack these encodings in the morphology matrix $X \in \mathbb{R}^{m \times n}$. A long short-term memory (LSTM) [9] embeds $X$ in the latent vector $c$. Conditioned by the latent vector $c$, a mapper predicts a capability measure $s$ for query position $p$. The learnable parameters of the encoder and the mapper are given by $\theta_1$ and $\theta_2$, respectively. The mapper is based on the work of Mescheder et al. [18]. A CResNetBlock is a pre-activation ResNet block [8] equipped with two conditional batch-normalization (CBN) layers [6]. The CBN parameters $\gamma_i$ and $\beta_i$ are predicted by fully connected layers with input $c$.

### B. Training

The prediction of a capability map for a manipulator requires accurate knowledge about its workspace. We account for this fact by first training a model with four CResNetblocks to predict binary reachability maps

$$
\tilde{f}(p) = \begin{cases} 1, & \text{if } p \in \mathcal{W} \\ 0, & \text{otherwise.} \end{cases} \tag{2}
$$

Afterward, we build a more sophisticated capability model on top of the pre-trained reachability model. To this end, we freeze the four trained CResNetBlocks and append an additional CResNetBlock.

For every training step, we randomly sample $n$ positions and capability measures $(p_i, \hat{s}_i)$ for a single robot from our training data and compute the loss as the mean-squared error

$$
\frac{1}{n} \sum_i (f_{(\theta_1, \theta_2)}(p_i, X) - \hat{s}_i)^2 . \tag{3}
$$

For the evaluation of model outputs $s_i$, we compute the coefficient of determination [5]

$$
R^2 = 1 - \frac{\sum_{i=1}^n (s_i - \hat{s}_i)^2}{\sum_{i=1}^n (s_i - \frac{1}{n} \sum_{j=1}^n \hat{s}_j)^2} . \tag{4}
$$

The advantage of this coefficient is that it considers both the absolute errors and the variance of the ground truth values. For a perfect model that accurately predicts the ground truth, $R^2 = 1$. If the model simply maps all inputs to the average of the ground truth, $R^2 = 0$.

## IV. DATA GENERATION

We use Timor-Python [13] to generate various manipulators from a set of robot modules. We combine five different types of modules – base (*B*), rotational joint (*J*), I-shaped link (*I*), L-shaped link (*L*), and end effector (*EEF*). While base, joint, and end effector modules are unique, we discretely parametrize the sizes of *I* and *L*-links. We sample the length of *I*-modules from $\{0.1, 0.2, 0.3, 0.4, 0.5\}$ and proximal and distal lengths of *L*-modules from $\{(0.1, 0.3), (0.5, 0.5), (0.8, 0.4)\}$. Every module is encoded by a seven-dimensional vector, where we use a one-hot index vector to encode the five different module types and concatenate it with two values indicating the module size parameters. To generate a robot, we randomly sample sequences of modules while ensuring that there are at most two links between two consecutive joints.

Next, we collect workspace data for every randomly generated robot. We repeatedly sample random joint angles and compute the corresponding end effector poses in the workspace (see Figure 1). In the second step, we find a tight over-approximation of the workspace $\overline{\mathcal{W}} \supset \mathcal{W}$ and discretize it. For every position in $\overline{\mathcal{W}}$, we compute the distance to the closest of the previously generated samples; if it exceeds a threshold distance $d_t$, we use a numeric inverse kinematics solver to check if it can be reached. If the inverse kinematics solver converges, we compute the corresponding capability measure, otherwise, we set it to the special value $s = \nu$. Finally, we add the new positions to the initial dataset.

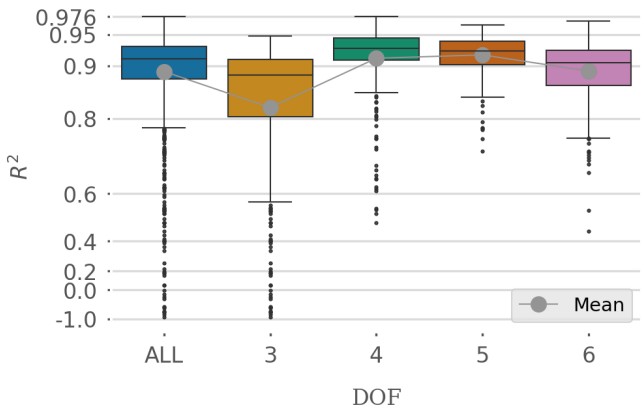

Fig. 3. Evaluation results of 1,239 test robots: The blue box plot presents the overall results, and the other boxes demonstrate the results according to the number of joints. The sample numbers in each group are 247, 263, 315, and 414 corresponding to the number of joints from three to six.

## V. NUMERICAL EXPERIMENTS

We generate a training set with 6,162 robots and a test set with 1,239 robots. To account for the inherent complexity of robots with more degrees of freedom, the distribution of robots with three, four, five, and six joints is $2 : 2 : 3 : 4$ in both sets, respectively. We use the translational manipulability index [29] as our capability measure for reachable positions. To distinguish singularities with a manipulability index of zero from unreachable points, we set $\nu = -1$.

Figure 3 shows the overall distribution of $R^2$ for the test set. The mean $R^2$ over all test samples is $0.8734$. For robots with three to six degrees of freedom (DOF), the mean $R^2$ is $0.7588, 0.9067, 0.9169,$ and $0.8873$, respectively.

Figure 4 visualizes slices of the 3D capability maps for four samples selected from the test set. These results visually underline the results from Figure 3, showcasing how the model is able to capture different workspace shapes and capability distributions. The inference time is $0.035$ seconds for more than $350,000$ query positions.

However, the evaluation results presented Figure 3 also show that there are still a few outliers. After manual inspection, we hypothesize that this can be attributed to capability maps with a low variance of manipulability indices, such as for manipulators with a planar workspace. Nevertheless, only $0.97\%$ of the test samples have an $R^2$ below zero.

Besides the test samples, we also evaluate the model on several out-of-distribution samples. Figure 5 shows the resulting maps for three variations of the six-DOF robot demonstrated in Figure 4. The first robot contains modules with new link lengths not present during training, while the following two robots consist of module chains not included in the generated dataset. For the first sample, we set the link size to $0.25$ for the $I$-link and the proximal and distal lengths to $0.12$ and $0.48$, respectively, for the $L$-link. For the second one, we switch the first joint-link pair (J,I) in the chain with a novel combination (J,I,L,I) consisting of a joint and three links. Finally, for the

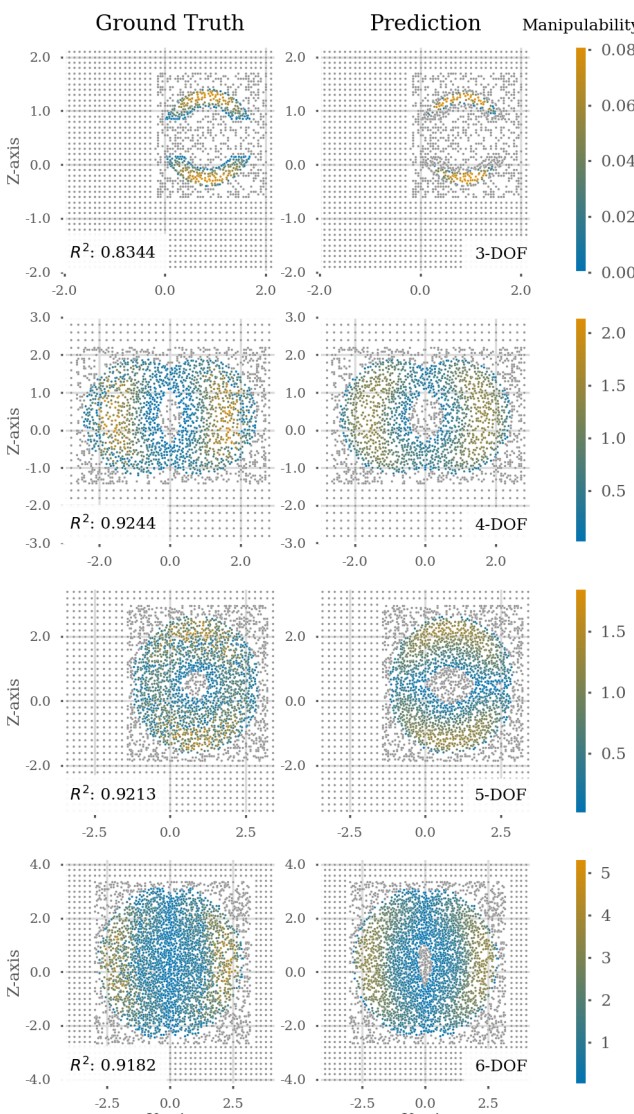

Fig. 4. Slices of capability maps with $x \in [-0.03, 0.03]$ for four different robots. The left side shows the ground truth, and the right side shows the generated capability maps. The color of a point indicates the corresponding manipulability measure, and grey points indicate positions labeled as "unreachable".

last sample, we add an additional joint to the beginning of the kinematic chain, creating a seven-DOF robot.

The first sample yields a high $R^2$ of $0.8589$ despite consisting of unfamiliar I-links and L-links. For the two samples with novel module chains, the model achieves an $R^2$ value of $0.8536$ and $0.8515$, respectively. These results indicate that our approach can extrapolate to manipulators that significantly differ from those it is trained on. This includes robots with more degrees of freedom than those in the training set.

## VI. CONCLUSION AND FUTURE WORK

We have developed an approach based on neural fields to efficiently generate high-quality capability maps of diverse robotic manipulators. Our experiments show that the trained

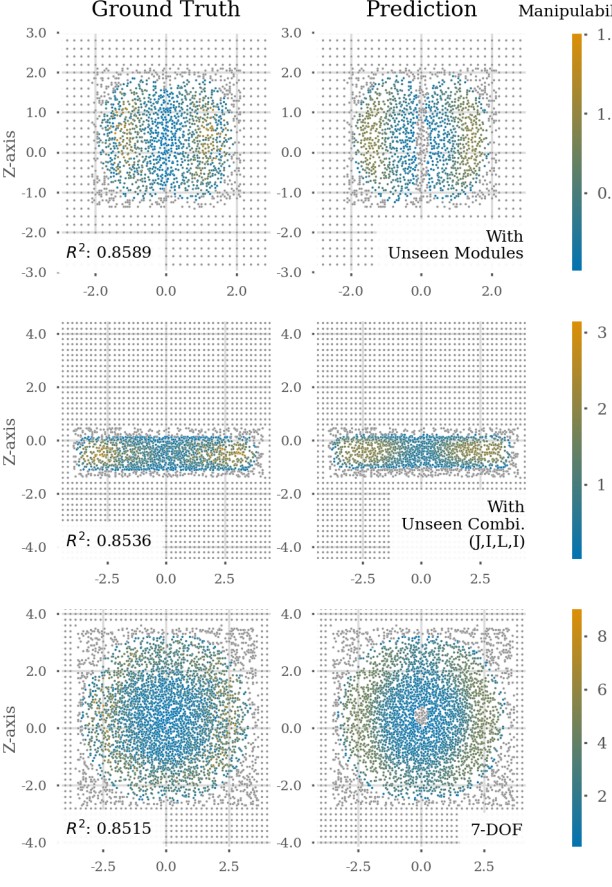

Fig. 5. Visualization of capability maps for out-of-distribution samples: We create three variations of the six-joint sample shown in Figure 4. The first one corresponds to robots consisting of previously unseen modules, and the next two correspond to previously unseen assemblies of modules. The left side shows the ground truth, and the right side shows the generated capability maps. The color of a point indicates the corresponding manipulability measure, and grey points indicate positions labeled as "unreachable".

model learns to accurately predict robot capability maps and can also extrapolate to kinematics built from module sequences not seen during training. Our approach works for arbitrary capability measures, such as the global measure proposed by Leibrandt et al. [14] and is only limited by the data that can be generated. A current limitation of our approach is that we currently do not account for self-collisions in the data generation process. In future work, we will extend the approach to workspaces in $SE(3)$, i.e., considering the manipulator orientation. Besides, the model could be combined with other algorithms, such as the work of Caverly et al. [3], to enable online self-calibration.

## ACKNOWLEDGMENTS

The authors gratefully acknowledge financial support by the Horizon 2020 EU Framework Project CONCERT under grant 101016007.

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
