# OpenReview forum: "Generating Robot Capability Maps with Neural Fields"
_roboticsfoundation.org/RSS/2024/Workshop/EARL — EARL 2024 Poster_

### Official Review · Reviewer_mqyG · 2024-06-18
**An interesting new method for generating robot capability maps with neural fields**

**Rating:** 8
**Confidence:** 3

**Review:**

## Summary:
This paper introduces a method to generate robot capability maps using neural fields, significantly reducing computation time from hours to milliseconds. The approach, trained on synthetically generated data, can generalize to diverse robot morphologies. The authors achieve an average R-squared score of 0.8734, demonstrating the method's accuracy and efficiency.

## Key Contributions:
1. **Efficient Map Generation:** Reduced computation time for generating capability maps.
2. **Generalization:** Strong generalization to unseen robot morphologies.
3. **Neural Fields Application:** Innovative use of neural fields in robot capability mapping.

## Methodology:
The paper presents a two-part methodology: data generation using Timor-Python to create diverse modular reconfigurable robots (MRRs) and a learning pipeline with an LSTM encoder and a mapper network to generate capability maps.

## Experimental Results:
The results show high accuracy (average R-squared 0.8734) and robustness, with effective generalization to out-of-distribution samples.

## Conclusion and Future Work:
The paper concludes with a promising outlook on the neural field-based approach, suggesting future work to consider manipulator orientation and self-collision in the reachability analysis.

## Strengths:
- Significant reduction in computation time.
- Strong generalization capabilities.
- Novel application of neural fields.

## Weaknesses:
- Current approach does not consider self-collision.
- No support of SE(3) workspace yet.

# Contextualization within EARL Workshop Agenda

The paper aligns with the themes of the EARL Workshop at RSS 2024, which focuses on the co-design of robot embodiment and control algorithms. Key areas of alignment include:
1. **Co-adaptation of Robot Design and Behavior:** Supports adaptive design and behavior integration.
2. **Generative Models for Robot Design:** Aligns with the use of generative models for optimizing robotic components.
3. **Scalable Co-optimization of Morphology and Control:** Directly supports scalable co-optimization methods.

Overall, the paper complements the workshop's goals by providing a method that enhances the efficiency and adaptability of robot capability analysis.

---

### Decision · Program_Chairs · 2024-06-24

Accept (Poster)